# Fifty years of change in the lower tree line in an arid coniferous forest in the Qilian Mountains, northwestern China

**Shu Fang** [1]*, **Zhibin He**[2], **Minmin Zhao**[3]

**1** College of Urban, Rural Planning and Architectural Engineering, Shangluo University, Shangluo, Shanxi, China, **2** Linze Inland River Basin Research Station, Chinese Ecosystem Research Network, Key Laboratory of Eco-hydrology of Inland River Basin, Northwest Institute of Eco-Environment and Resources, Chinese Academy of Sciences, Lanzhou, Gansu, China, **3** Key Laboratory of Hydrogeology, Center for Hydrogeology and Environmental Geology Survey, China Geological Survey, Baoding, Hebei, China

☯ These authors contributed equally to this work.
* 201930@slxy.edu.cn

**Data Availability Statement:** All relevant data are within the paper.

**Funding:** This research was funded by The Natural Science Special Program of Shaanxi Provincial Department of Education, No. 21JK0614; The PhD

## Abstract

Tree line areas exhibited significant changes in response to climate change, including upward migration. Lower tree line dynamics are rarely studied, but as unique features in arid and semi-arid areas, they may influence forest distribution. Here, eight lower tree line plots in a *Picea crassifolia* Kom. (Qinghai spruce) forest in the arid and semi-arid Qilian Mountains of northwestern China were used to determine changes in tree line location and relationships with meteorological factors during 1968–2018. The results showed that the lower tree line descended by an average of 9.82 m during 1968 to 2018, and exhibited almost no change after 2008. The change in the lower tree line was significantly correlated with the annual average temperature (˚C) and annual precipitation (mm) and may be affected by human activities. In the past 50 years, the lower tree line in arid areas exhibited a downward trend. Our findings indicate that the movement of the lower tree line is also an important aspect of climatic changes in coniferous forest distribution in arid and semi-arid mountains.

## Introduction

Tree lines are apparent vegetation boundaries, commonly defined as the elevational limit of trees greater than 2 m in height [1–4]. The area extent of tree lines has geological and ecological significance, and changes in tree lines play an important role in the development of mountains and regional ecology [5–7]. Over the past century, mountain areas have experienced greater climate warming, increased carbon dioxide concentration, and increased extreme weather events than the global average [8, 9]. Tree line ecotones are more sensitive to climate change and are therefore ideally suited for climate change monitoring [10–13]. The rise in temperature leads to the boundaries of global species at a rate of 6.1km/10a to the North and South, and in mountainous areas, it is manifested as the boundary between species; the phenomenon of the rising tree lines has been observed in Europe, North America, New Zealand, and other regions and, in some forests, where the tree line did not expand upward significantly, tree line areas experienced a gradual alteration from tundra to forest [14, 15].

early development program of Shangluo
University, No.19SKY027; The Innovation team of
water source protection of middle route of South-
to-North Water Diversion (SK2017-44), The
Shaanxi Ecohydrology Observation and Research
Station of the Southern Qinling Mountains and The
Cohydrology Observation and Research Station of
the Southern Qinling Mountains. The funders had
no role in study design, data collection and
analysis, decision to publish, or preparation of the
manuscript.

**Competing interests:** The authors have declared
that no competing interests exist.

Tree lines are comprised of the upper and lower forest boundaries, and they determine the distribution of forests in mountain regions. However, the lower tree lines generally only appear in arid or semi-arid mountainous areas worldwide [16, 17]. In the arid and semi-arid mountainous areas along the lower tree line, climate change is more complex than elsewhere [18] and exhibits increases in temperature, decreases in precipitation frequency, and increases in extreme rainfall events [19]. More significant climate change and the scarcity of water resources render tree line vegetation in arid areas more fragile and more susceptible to climate change [20].

Although the mountain environment where the lower tree lines are located is more fragile and sensitive, the lower tree lines received little scientific attention [21]. In contrast, much research has been conducted in the upper lines of mountain forest belts, including studies on distribution, formation mechanism, and the response of tree line regional location to climate change [8, 22–24]. Analyses of global forest lines at high latitudes and altitudes since 1900 showed that the location of 52% of the mountain tree lines has shifted upward [42]; however, the dynamics at the lower tree lines remain unclear, and research on the lower tree line in the arid and semi-arid areas in northwestern China is also relatively scarce [25]. In this study, we aimed to determine lower tree Line dynamics in the semi-arid and arid mountains which experience a significant climate change.

Many studies focused on the relationship between tree line position and meteorological factors. It has been determined that the change in the upper tree line is mainly related to temperature, while the change in the lower tree line is limited by precipitation. The formation of the upper tree line is limited by the growing season mean air temperature of 5.5 to 7.0 ˚C or a growing season mean soil temperature of 6.7–0.8 ˚C at 10 cm soil depth [26, 27]. Further, at lower tree line positions, tree demographic processes are more vulnerable to water stress and balance [28, 29]. Also, radial growth at lower tree lines appears more closely related to meteorological factors, in particular to precipitation from May and June [16]. The change in temperature controls the melting of snow and ice, and is coupled with the effects on precipitation, soil humidity, tree growth, and the interaction among plant species [30].

Located in the northwestern arid region of China, forest vegetation of the Qilian Mountains is not only a valuable forest resource, but also provides the ecological function of water conservation [31]. Under the influence of climate change in recent years, vegetation cover and growth in Qilian Mountains have changed significantly [32] and the cover of the main dominant and establishment tree species, *Picea crassifolia* Kom., has also increased, and the boundary of the upper tree line has been moving up [33]. Research on *Picea crassifolia* Kom. mostly focuses on dendroclimatology and changes in the upper tree line [34–36]. Thus, we chose *Picea crassifolia* Kom. forest lower tree line in the arid region of northwestern China to study the response of the lower tree line position to climate change.

We selected a typical catchment in the Qilian Mountains to conduct a sample survey of the lower tree line area to determine changes, if any, over the past 50 years: (1) the position of *Picea crassifolia* Kom. forest at the lower tree line, and (2) the main meteorological factors (average monthly temperature, monthly precipitation, seasonal average temperature, seasonal precipitation, annual average temperature, annual precipitation) affecting the changes in *Picea crassifolia* Kom. forest at the lower tree line. We aimed to increase the understanding of the processes at the lower tree line in arid and semi-arid regions undergoing climate change.

## Materials and methods

### Study area and method of field sampling

We conducted this study at the Dayekou (38˚26′-38˚35′N, 100˚14′-100˚19′E), the typical catchment in the Qilian Mountains of China [37, 38], at an altitude of 2500–4700 m. The

foundation species is *Picea crassifolia* Kom, the area accounts for 75.7% of the total area of the forest [39]. The *Picea crassifolia* Kom has patchy distribution on shaded and semi-shaded slopes at an altitude of 2500–3400 m; the area also contains sparse *Qilian juniper (Sabina prze-walskii Kom.)*. The soil is mainly mountain gray-brown and subalpine shrub meadow, and is characterized as relatively thin, with mainly silt sand texture [40]. The annual temperature in the Dayekou catchment ranges 2.0–3.5˚C, with July temperatures ranging from 10–14˚C. Annual precipitation ranges 350–450 mm and is concentrated in June-September. Annual evaporation is 1050–1100 mm and the annual total of sunshine hours is 1890–1900. The annual average relative humidity ranges from 60 to 65% [41].

A sample survey along an elevational transect can be used to reconstruct deterioration of tree conditions; it is a powerful tool to investigate the relationships of tree growth and climate change [23]. We established plots in Dayekou catchment that met the following conditions: 1. The soil conditions of all plots are consistent and belong to mountain gray brown. The lower tree line was obvious, and 2. plots were representative of the lower tree line conditions at different altitudes and slope aspects. A total of 8 plots, each 50 x 30 m, were surveyed between July 31 and Oct. 11, 2018 (Fig 1, Table 1). Plots were set up on the edge of *Picea crassifolia* Kom. lower tree line. We set the spline perpendicular to the slope direction and along the slope direction to ensure that one side is parallel to the maximum slope of the area and includes the boundary of the tree line. Tree height of 1.3 m meets the division of the tree line (> 2 m), and the measurement of DBH (diameter at breast height) (height of breast diameter is 1.3 m, [42]), so we recorded tree positions (x and y coordinates, defined with respect to the lower left corner of the plot), DBH (height of breast diameter is 1.3 m), of all trees with height > 1.3 m.

## Tree age assessment and the determination of the tree line position

Tree age assessment of each sample was based on the survey results of the *Picea crassifolia* Kom. forest sample plot established in the Guantai area (100.15˚E, 38.32˚N) in Dayekou catchment, which is very close to our research area. This plot was established in a monospecific

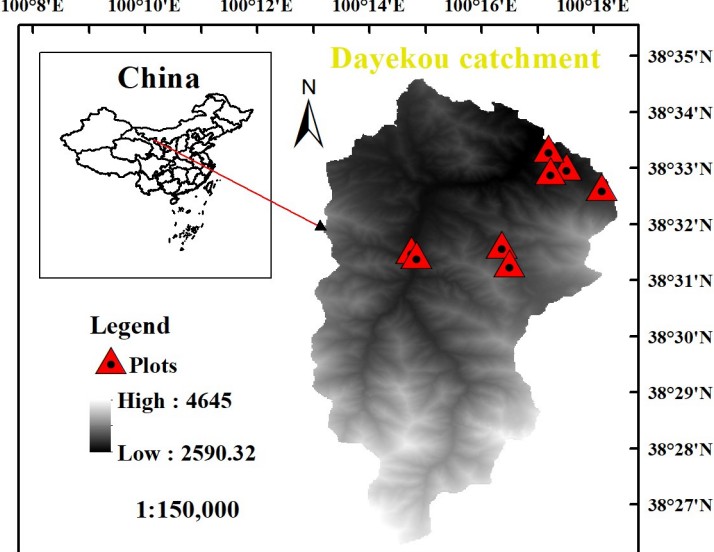

**Fig 1. Study area in the Qilian Mountains of northwestern China.**

**Table 1. Characteristics of the sites, in aspect, 1 for shaded slopes (N (0–22.5˚, 337.5–360˚)), 2 for semi-shaded (NE (22.5–67.5˚), E (67.5–112.5˚), NW (292.5–337.5˚)), and 3 for partly-sunny slopes (SW (112.5–157.5˚), SE (202.5–247.5˚), and W (247.5–292.5˚)).**

| Plot Number | Latitude | Longitude | Elevation (m) | Aspect | *Picea crassifolia* Kom. (height > 1.3 m) number | *Sabina przewalskii* Kom. (height > 1.3 m) number |
|---|---|---|---|---|---|---|
| 1 | 100.29˚E | 38.62˚N | 2595 | 2 | 156 | 0 |
| 2 | 100.30˚E | 38.54˚N | 2980 | 2 | 397 | 10 |
| 3 | 100.29˚E | 38.55˚N | 2800 | 1 | 282 | 0 |
| 4 | 100.27˚E | 38.53˚N | 3013 | 2 | 176 | 1 |
| 5 | 100.25˚E | 38.53˚N | 2852 | 3 | 75 | 68 |
| 6 | 100.25˚E | 38.52˚N | 2848 | 2 | 112 | 25 |
| 7 | 100.27˚E | 38.52˚N | 3091 | 1 | 61 | 0 |
| 8 | 100.29˚E | 38.56˚N | 2673 | 2 | 170 | 1 |

*Picea crassifolia* Kom. forest to study basic characteristics; in this plot, we obtained tree cores (at 1.3 m) and measured DBH (at 1.3 m) of 358 trees with a height of > 1m. Trees in the sample plot cover *Picea crassifolia* Kom. forests of various ages, and the age of each tree was determined with tree core sampling that according to the standard annual ring technology, use growth cones to drill the core of the tree cross section at a height of 1.3 m, and ensure that at least one specimen of each tree hits the core [42]. Based on the data from the Guantai site, we first calculated the average DBH of each tree age group. Then, we fitted a function between DBH and the age of the tree. Then we counted and calculated the average proportion of *Picea crassifolia* Kom. tree at different ages in 8 plots.

The field sampling site is located in the natural protection of the west water in the middle of the Qilian Mountain National Nature Reserve in Gansu Province, so we assume that nothing happened to all permanent plots during that period [39]. To determine the rate of tree line shift from 1968 to 2018, we defined the lower tree line position for an individual slope as the mean elevation of the 20 lowermost *P. crassifolia* trees>2 m in height that were alive in a given year [3, 33]. Then, using this definition, we determined the elevation (y coordinate) of lower tree line of *Picea crassifolia* Kom. forest in different years (1968, 1978, 1988, 1998, 2008, and 2018). Then we calculated the average elevation of the lower forest line in different periods.

## Correlation analysis between lower tree line and climate change variables

We used climate data from the nearest meteorological station at Yeniugou located at a mean distance of < 45 km and altitude of 3180 m. We calculated climate variables of average monthly temperature, monthly precipitation, seasonal average temperature, seasonal precipitation, annual average temperature, annual precipitation from 1968–2018 to determine the relationship between *Picea crassifolia* Kom. lower tree line change and meteorological factors.

To study the impacts of climate change on tree recruitment in the lower tree line, we used Pearson's correlation coefficient between the average value of different meteorological factors in each decade and the mean distance of descent of the lower tree line tree between years 1968 and 2018 using SPSS 19. The meteorological factors we selected were annual average temperature (˚C), annual precipitation (mm), spring average temperature (˚C), summer average temperature (˚C), autumn average temperature (˚C), winter average temperature (˚C), spring precipitation (mm), summer precipitation (mm), autumn precipitation (mm), and winter precipitation (mm). Significance level was P = 0.01 or P = 0.05. If the sample's correlation coefficient is greater than or equal to this critical value, the correlation passes the test.

## Results

### The characteristics of *Picea crassifolia* Kom. lower tree lines

We calculated stand density, measured average DBH, and determined forest age of the plots at the lower tree line (Table 2). We found that the average number of *Picea crassifolia* Kom. trees on shaded and semi-shaded slopes was about 200, while that on the partly-sunny slope was only 75; the number of *Sabina przewalskii* Kom. on the partly-sunny slope was almost the same as that of *Picea crassifolia* trees. Stand density was less than 0.1 trees/m$^2$ in plots at an altitude > 2800 m. Stand density was negatively correlated with average DBH (r = -0.720, P<0.05) and average forest age (r = -0.721, P<0.05). The smaller the stand density, the larger the average DBH and average tree age. Canopy and tree line density at the lower tree line at an altitude below 2600 m and above 3000 m were slightly lower than those in other tree line plots.

### Dynamic changes in tree line position of the lower tree line

The simulation found that the power function (compared with linear function, quadratic function, exponential function, logarithmic function) had the best fit between DBH and the age of the tree. The formula was $Y = 4.8827X^{0.9661}$ ($R^2 = 0.7253$, P <0.001) (Fig 2). Using this simulation formula, we calculated the age of all *Picea crassifolia* Kom. forests in the sample based on the DBH data of each tree in the sample survey. The average proportion of trees at different ages at the lower tree line was similar at all sampling sites. The younger trees that have been recruited after 1980 reached 68% of the current population; and reached 78% since 1960 (Fig 3).

The change in the lower tree line position in plots is shown in Fig 4. Plot 1 did not have a tree line in 1968 and tree line descended by 8.11 m from 1978 to 2018; the rate of descent has been decreasing from 1978 to 2018. The tree line for plot 2, 3, and 4 descended by 23.55, 18.85, and 10.20 m, respectively, between 1968 and 2018. In plot 3, the rate of tree line descent first decreased, and then increased slightly from 1998 to 2008; while in plot 2 and 4, it showed a slight upward trend from 1968 to 1998, and then fell sharply. The descent of the tree line in plots 5–8 was relatively slow than in plots 1–4, moving down by 3.54, 2.58, 6.36, and 5.41 m, respectively, between 1968 and 2018. The decline rate in plots 5, 7, and 8 first decreased and increased during 1988–1998, and then rapidly decreased. The descent rate of the tree line in plot 6 slowed in 1998–2008, and then stabilized.

The lower tree line has moved down by an average of 9.82 m in the last 50 years (Table 3). The decline was greatest in 1988–1998. Although the descent continued after 1998, the extent was much smaller and the lower tree line remained almost unchanged from 2008 to 2018.

**Table 2. Survey characteristics of the sites.**

| Plot Number | Stand density (number/m$^2$) | Average DBH (cm) | Standard deviation of DBH (cm) | Average forest age (year) | Standard deviation of forest age (year) |
|---|---|---|---|---|---|
| 1 | 0.1 | 5.4 | 3.6 | 25 | 16 |
| 2 | 0.26 | 6 | 6.8 | 27 | 29 |
| 3 | 0.19 | 9.5 | 7.1 | 43 | 31 |
| 4 | 0.12 | 8.5 | 6.6 | 38 | 29 |
| 5 | 0.05 | 18 | 11.9 | 79 | 51 |
| 6 | 0.07 | 13.2 | 12.1 | 58 | 52 |
| 7 | 0.04 | 16.3 | 12.8 | 72 | 55 |
| 8 | 0.11 | 12.3 | 7.2 | 55 | 31 |

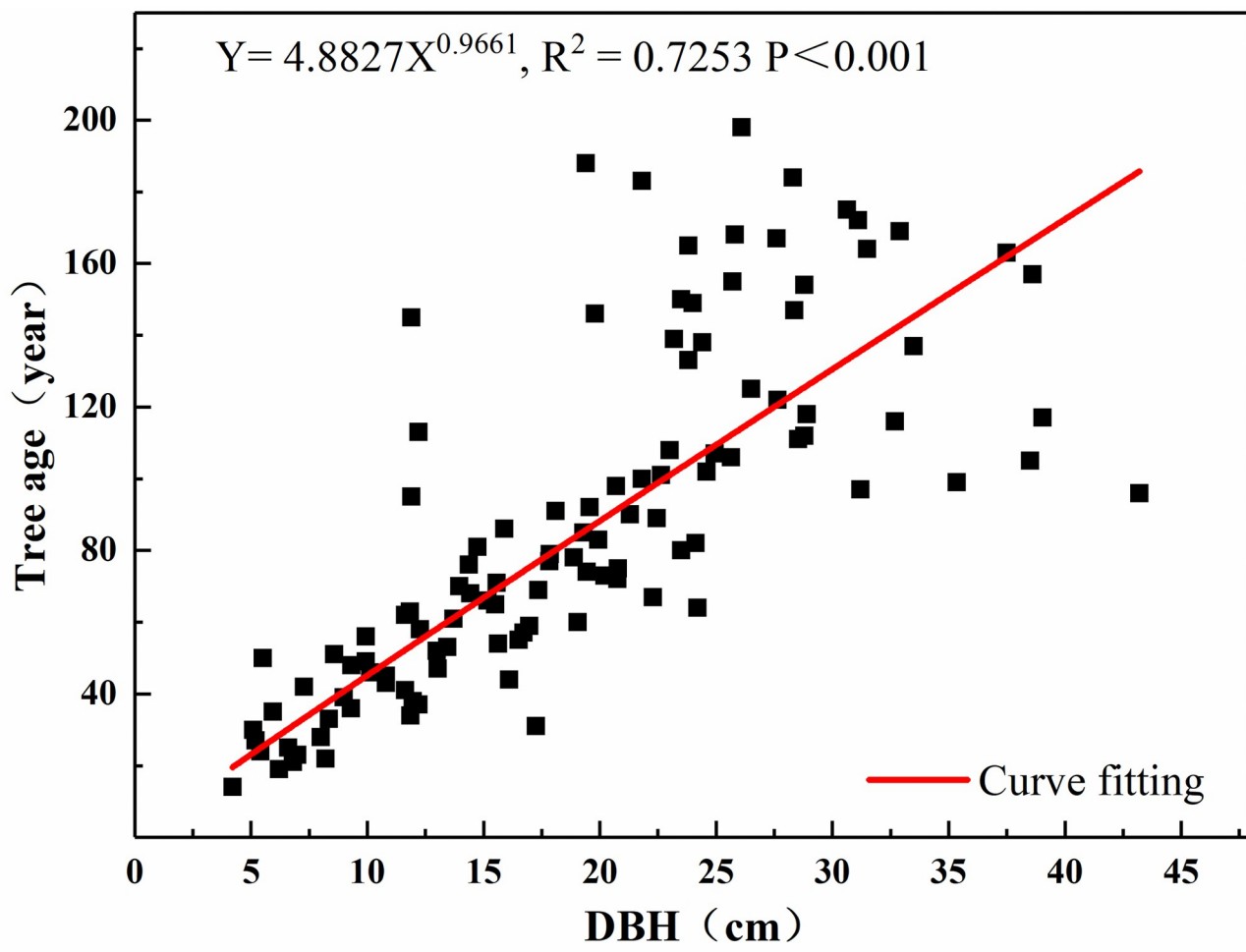

**Fig 2. Functional relationship between DBH and age of trees in Guantai sampling.**

### Relationship between tree line changes and meteorological factors

We calculated climate trends at the Yeniugou weather station, and determined that the temperature increased at a mean rate of 0.41°C per decade from 1968 to 2008, and there is a slight drop after 2008. The annual average temperature in the region increased by a total of 2.00°C from 1968 to 1998 but decreased by a total of 0.49°C from 1998 to 2018. Annual precipitation increased with a mean rate of 0.82 mm per year and the change in annual precipitation was mainly caused by increased summer precipitation.

The correlation between the average value of different meteorological factors in each decade and mean distance of descent of the lower tree line between 1968 and 2018 is shown in Table 4. The results show that the change in the lower tree line was significantly correlated with the annual average temperature (°C) (r = 0.907, P = 0.033) and annual precipitation (mm) (r = 0.911, P = 0.031). The correlation between the degree of decrease in the tree line and the change in annual precipitation was the highest. In addition, the elevation of tree line decline was positively correlated with the change in summer temperature and the change in winter precipitation, with the correlation coefficient > 0.5. The elevation of decline in the tree line was negatively correlated with the changes in other meteorological factors, with the correlation coefficient of < 0.5.

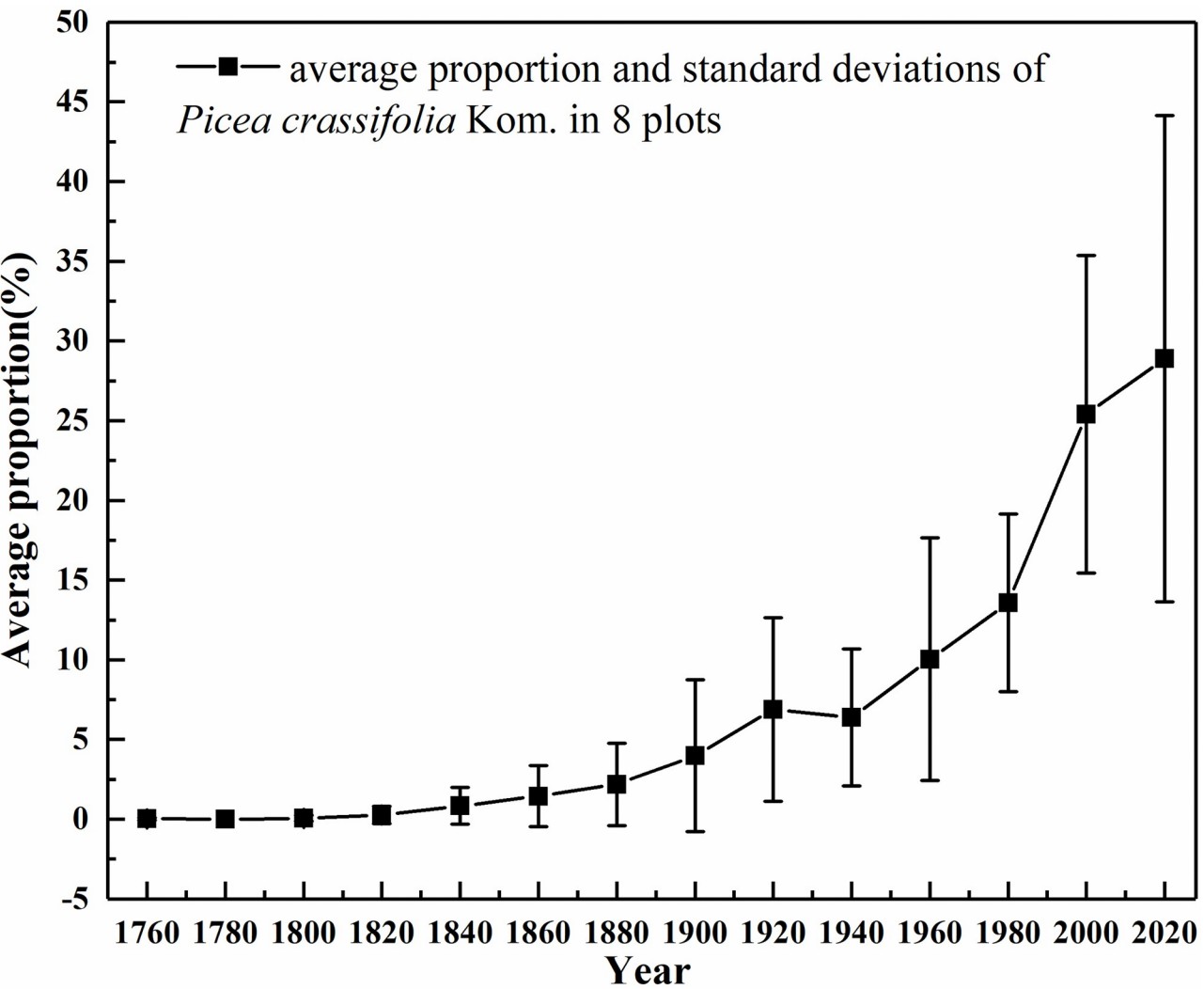

**Fig 3. The average proportion of Picea crassifolia Kom. trees at different ages in 8 plots.**

Analysis of meteorological factors and changes in the location of the lower tree line showed that the trend in temperature warming in years 1968 to 2008 has been weakening, especially in 1998–2008 (the increase in temperature was < 1°C), and the lower tree line descent between 1968 and 2008 also slowed. There was a slight decrease in temperature after 2008, and the elevation of the lower tree line remained unchanged. Similarly, the trend of increasing precipitation in 2008–2018 has also weakened.

## Discussion

We set up plots in a typical *Picea crassifolia* Kom. catchment in the Qilian Mountains to determine changes in the location of the lower tree line and their relationship with climate change.

### Variability in lower tree line position

The lower tree line decreased by an average of 9.82 m from 1968 to 2018, with the *Picea crassifolia* Kom. tree line ecotone exhibiting a downward tendency to expand. An advance of the

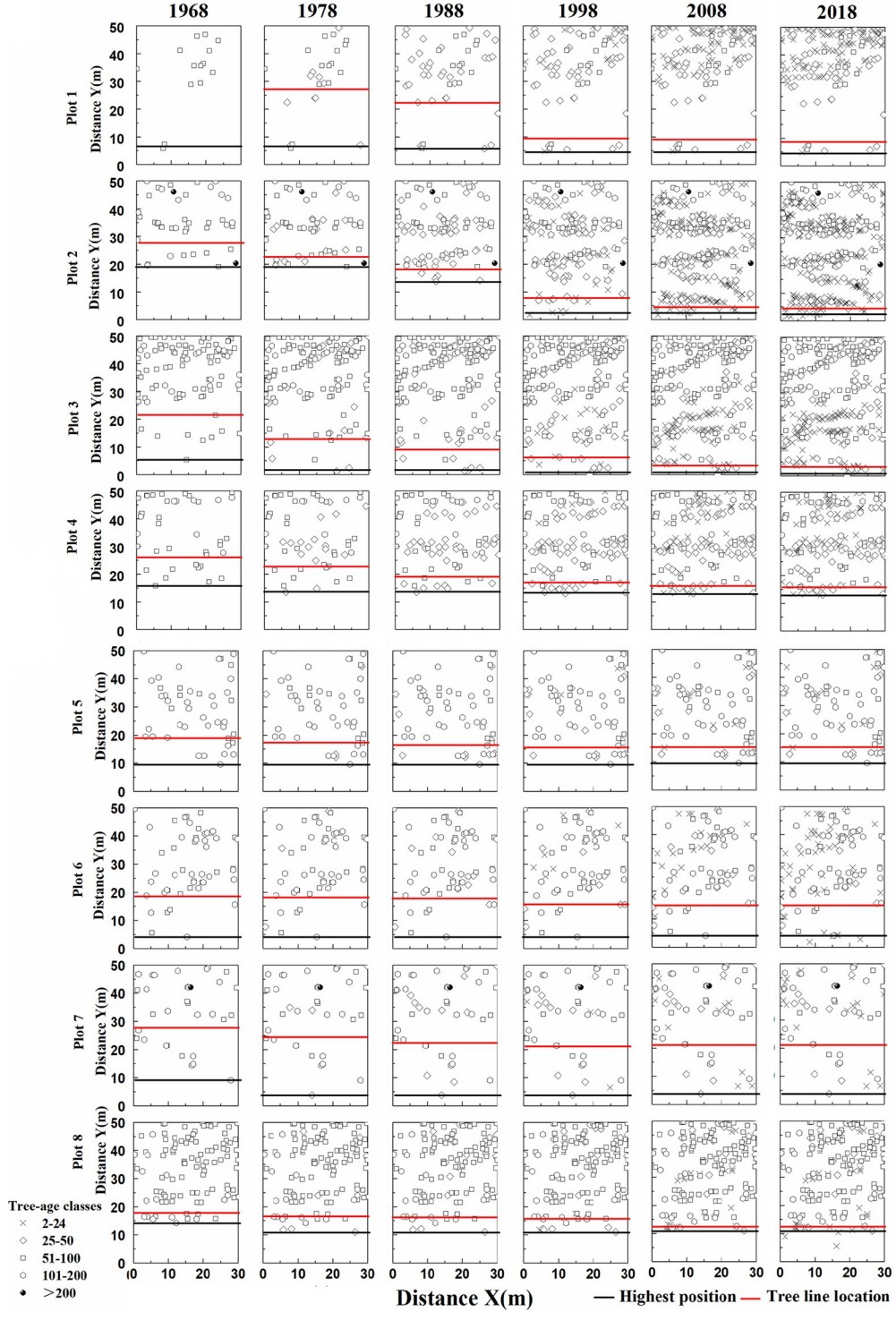

**Fig 4. Changes in position of the lower tree line (1) (distance Y(m) is the distance along the slope direction and distance X(m) is the distance spline perpendicular to the slope direction).**

**Table 3. Decadal and 50-year mean distance and standard deviation of descent elevation of the lower tree line.**

| Year | 1968–1978 | 1978–1988 | 1988–1998 | 1998–2008 | 2008–2018 | 1968–2018 |
|---|---|---|---|---|---|---|
| Plot 1 (m) | | 5.22 | 2.77 | 0.12 | 0.01 | 8.11 |
| Plot 2 (m) | 3.86 | 4.98 | 11.54 | 3.19 | 0.00 | 23.55 |
| Plot 3 (m) | 8.00 | 3.88 | 3.26 | 3.72 | 0.00 | 18.85 |
| Plot 4 (m) | 2.86 | 3.19 | 3.30 | 0.84 | 0.00 | 10.20 |
| Plot 5 (m) | 1.28 | 1.85 | 0.07 | 0.36 | 0.00 | 3.54 |
| Plot 6 (m) | 0.84 | 0.60 | 0.47 | 0.67 | 0.00 | 2.58 |
| Plot 7 (m) | 2.35 | 2.65 | 1.36 | 0.00 | 0.00 | 6.36 |
| Plot 8 (m) | 1.67 | 0.30 | 1.62 | 1.83 | 0.00 | 5.41 |
| Mean distance of descent elevation of the lower tree line (m) | 2.98 | 2.83 | 3.05 | 1.34 | 0.00 | 9.82 |
| Standard Deviation (m) | 2.44 | 1.85 | 3.64 | 1.43 | 0.00 | 7.53 |

tree line has been observed in Europe, North America, New Zealand, and in other regions worldwide [14, 15]. In some forests where the tree line did not expand upward significantly, such as in the subalpine fir forests in Glacier National Park, USA, tree line areas experienced a gradual alteration from tundra to forest [43]. At high altitudes and latitudes, the tendency of the tree line area to expand has been detected and attributed to increasing temperatures [44–46]. However, most of the research focused on the advancement of the upper tree line that 52% of the mountain tree line positions have shifted upward [42] while the dynamics at the lower tree line remain unclear. In our study, the lower tree line descended by an average of 9.82 m between 1968 and 2018. Forest cover increased in the Swiss Alps between 1909 and 2009 at the highest and lowest elevations for all slope aspects [47]. Similarly, a remarkable increase in density and an upslope shift of 50 m was observed between 1912 and 2009 at the lower boundary of the tree line ecotone in sub-Arctic Sweden; the main factor was human impact of decreasing forest area [48].

In the Qilian Mountains, vegetation cover appears to have been increasing during 1982–2014 [49]. In site survey, 78% of the trees were recruited after 1960, therefore, most of the trees were small and with an increase in tree age, the landscape pattern gradually changed from cluster to random distribution [50]. Carbon mass of *Picea crassifolia* Kom. forest in the Qilian Mountains has increased by 1.202 kg/m$^2$ from 1964 to 2013 [41]. The correlation analysis between the change in the lower tree line position and meteorological factors showed that the change in the lower tree line was highly correlated with annual precipitation and annual average temperature; the rate of change in these two factors after 2008 has slowed down, so the elevation of decline in the lower tree line has also slowed down. At the same time, the change in trees in the tree line area may be from expansion of the forest to increase in tree density. During adaptation to climate change, forest ecosystems may fill in their interiors rather than shift the position of the tree line [51–53]. In a study of the forest

**Table 4. The relationship between mean distance of descent elevation of the lower tree line (m) and different meteorological factors from 1968–2018.** (M represents mean distance of descent elevation of the lower tree line (m), C1 to C10 represent annual average temperature (˚C), annual precipitation (mm), spring average temperature (˚C), summer average temperature (˚C), autumn average temperature (˚C), winter average temperature (˚C), spring precipitation (mm), summer precipitation (mm), autumn precipitation (mm), winter precipitation (mm); * represents a significant correlation at the 0.05 level (bilateral)).

| Correlation | | C1 | C2 | C3 | C4 | C5 | C6 | C7 | C8 | C9 | C10 |
|---|---|---|---|---|---|---|---|---|---|---|---|
| M | R | 0.907* | 0.911* | -0.352 | 0.703 | -0.147 | -0.243 | -0.436 | -0.288 | -0.190 | 0.683 |
| | P | 0.033 | 0.031 | 0.562 | 0.186 | 0.814 | 0.694 | 0.463 | 0.638 | 0.760 | 0.204 |

line in the Himalayan Mountains (including the Qilian Mountains), the changes in forest line exhibited high heterogeneity, with 67.7% of the tree lines shifting upward, 73% of the forest lines exhibiting an increase in tree recruitment, and 67.8% of the forest lines presenting acceleration of tree growth rates [4].

One survey showed that *Picea crassifolia* Kom. forest density increased 23-fold at the tree line, but the tree line position was not significantly altered during the past 100 years [54]. Another survey indicated that the elevation of the upper tree line shifted upward by 6.1 to 10.4 m between 1957 and 1980, but showed no obvious changes in years 1980 to 2007 [33]. Further, the number, area, and concentration of forest patches increased in years 1968 to 2017 in relatively flat and partly-sunny areas, but the rate of area increase and ascent of the tree line slowed after the year 2008 [55].

Our results were similar to those of others from the Qilian Mountains; the lower tree line in the Qilian Mountains has shifted downslope in the past 50 years by an average of 9.82 m, and the expansion rate has begun to decline after 1998, with no change after 2008.

## Meteorological factors affecting the change of lower tree line

Meteorological factors affect the distribution of tree lines and tree regeneration rates [3]. In our study, the altitudinal variation of *Picea crassifolia* Kom. forest at the lower tree line position was correlated with annual precipitation and annual average temperature. At high altitudes and latitudes, the tendency of tree line area to expand had been detected and attributed to the warming temperature [44–46]. The main meteorological factor that affects the distribution of the alpine tree line in China is the temperature during the growing season, and mostly precipitation affects the distribution of the tree line indirectly through temperature [56]. Recent studies have shown that precipitation has a comparable or stronger effect on the position of tree line than temperature, as enough precipitation and moisture may promote the growth rate of trees and the establishment of trees [57, 58].

For *Picea crassifolia* Kom. forests that grow in high-altitude areas with sufficient rainfall, June temperatures control the melting of snow and ice, and July-August temperatures affect radial growth [30], and average temperature in the wettest season (growth season) has a significant effect on *Picea crassifolia* Kom. forests growth [20]. The expansion and regeneration at the lower tree line is also limited by annual rainfall [17]. A study which combined remote sensing and plot surveys in *Picea crassifolia* Kom. forests in Dayekou catchment revealed that the upper tree line was distributed at the threshold of mean annual air temperature at the upper elevation boundary of −2.59 to −2.73°C, while the lower tree line was distributed at the threshold of mean annual precipitation of 378.1 to 372.3 mm [59].

## Uncertainty discussion

Tree line changes are complex and variable, with significant regional differences; therefore the reasons for such changes between the forest and the open ecosystem are multiple and still part of the debate.

The annual mean temperature and annual precipitation are only two of several other meteorological factors that affect the change in the lower tree line. Our correlation analysis indicated that the summer temperature also had a positive correlation with the rate of descent of the forest line. The expansion of the lower tree line was related to the decrease in low temperatures or frosts damaging tree seedlings caused by the increase in summer minimum temperature [60]. Lower nighttime minimum temperatures and short continuous frost-free periods may prevent the establishment of tree seedlings in the lower tree line through photoinhibition,

tissue damage or frost heaving; fine soil may also hinder the establishment of tree seedlings in the lower tree line [61].

Topographic factors affecting local environmental conditions such as slope, can also affect the development of vegetation in the lower tree line [62]. A study in a semi-arid area of Greece from 1945 to 2015 showed that changes in temperature and precipitation only have the theoretically achievable elevation of the tree line, the main factor determining whether the trees in the tree line area expanded was the difference in wind caused by topography [63]. The main factors responsible for the distribution of *Eucalyptus coccifera* lower tree line on the Mt. Wellington Plateau, Tasmania, were soil drainage characteristics [64].

Changes in the development of vegetation in the lower tree line are affected not only by periodic climate fluctuations and microenvironment caused by topography, but also by disturbances, especially the role of humans in fire frequency, forest coverage, or composition [65]. In the Himalayas, human activity was one of the reason why the trees have not reached the height of the forest line height of the forest line that is limited by the average temperature [66]. Before Qilian Mountain was identified as a national natural area in 1988, it was affected by human activities such as thinning [67]. Therefore, the decline of tree line may be the illusion of the decline of tree line elevation caused by the destruction of old trees.

In addition, the position changes of the lower tree line in arid areas may also be due to the influence of tree regeneration, time lag of climate adaptation, and the disappearance of permafrost [68].

It is expected that the tree line in the eastern Himalayas will continue to expand in the end of the 21st century, resulting in the loss of at least 20% of its existing habitat for endemic plants [69]. Our study shows that, in addition to considering the increase of the elevation of the tree line, it is also necessary to consider the changes in the relationship between alpine plants brought about by the downward expansion of the tree line.

*Picea crassifolia* Kom. grows slowly, so we chose ten years as the time scale for studying tree line change. Then, to explore the relationship between tree line change and meteorological factors, the correlation analysis was done at that time scale, too. Therefore, the amount of data is small, and analysis of the relationship between tree line changes and meteorological factors supported the discussion on the possibility of tree line changes only. In follow-up work, tree rings will be considered to more accurately establish the relationship between tree growth and climate change. The growth of trees in the tree line area will gradually reduce the dependence on temperature increases, due in turn to local factors other than climate warming [70]. Studies have shown using climate change was not a robust explanation of changes in the location of tree lines. More factors should be introduced to explore changes in forest lines so as to formulate more appropriate policies to cope with climate change.

## Conclusions

The lower tree line of *Picea crassifolia* Kom. forest exhibited a descending trend from 1968 to 2008, and almost no change after 2008. The change in the lower tree line was highly correlated with temperature, and related to annual precipitation. In the past 50 years, the lower tree line of *Picea crassifolia* Kom. montane forests in this study exhibited a downward trend under climatic conditions of higher temperatures and increased precipitation. We will combine high-definition remote sensing data to further explore the mechanism of tree line changes in arid and semi-arid areas.

## Acknowledgments

We are very grateful to Kathryn Piatek for her comments and editorial assistance.

## Author Contributions

**Data curation:** Shu Fang.

**Funding acquisition:** Shu Fang.

**Investigation:** Zhibin He.

**Methodology:** Shu Fang, Zhibin He.

**Software:** Shu Fang, Minmin Zhao.

**Writing – review & editing:** Minmin Zhao.

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
