## [Decision Letter · Decision Letter 0]

18 Jul 2023

PONE-D-23-19267

Fifty years of change in the lower tree line in an arid coniferous forest in the Qilian Mountains, northwestern China

PLOS ONE

Dear Dr. fang,

Thank you for submitting your manuscript to PLOS ONE. After careful consideration, we feel that it has merit but does not fully meet PLOS ONE’s publication criteria as it currently stands. Therefore, we invite you to submit a revised version of the manuscript that addresses the points raised during the review process.

 Both reviewers are concerned about the exclusion of trees less than 1.3m height, which will cause the uncertainty of the final conclusions. I agree. In addition, one of the reviewers argued that forests in the lower elevational sites in Qilian Mountain are seriously affected by human activity. In other words, human activity would cause uncertainty of the conclusions too. So, I suggested the authors make a major revision based on these comments.

We look forward to receiving your revised manuscript.

Kind regards,

Bao Yang, Ph.D, Prof.

Academic Editor

PLOS ONE

Journal Requirements:

   "This research was funded by The Natural Science Special Program of Shaanxi Provincial Department of Education, No. 21JK0614 ; The PhD early development program of Shangluo University, No.19SKY027; The Innovation team of water source protection of middle route of South-to-North Water Diversion (SK2017-44), The Shaanxi Ecohydrology Observation and Research Station of the Southern Qinling Mountains and The Cohydrology Observation and Research Station of the Southern Qinling Mountains."

6. We note that Figure 1 in your submission contain map/satellite images which may be copyrighted. All PLOS content is published under the Creative Commons Attribution License (CC BY 4.0), which means that the manuscript, images, and Supporting Information files will be freely available online, and any third party is permitted to access, download, copy, distribute, and use these materials in any way, even commercially, with proper attribution. For these reasons, we cannot publish previously copyrighted maps or satellite images created using proprietary data, such as Google software (Google Maps, Street View, and Earth). For more information, see our copyright guidelines: http://journals.plos.org/plosone/s/licenses-and-copyright.

Additional Editor Comments:

Both reviewers are concerned about the exclusion of trees less than 1.3m height, which will cause the uncertainty of the final conclusions. I agree. In addition, one of the reviewers argued that forests in the lower elevational sites in Qilian Mountain are seriously affected by human activity. In other words, human activity would cause uncertainty of the conclusions too. So, I suggested the authors make a major revision in response to these comments carefully.

Reviewers' comments:

Reviewer's Responses to Questions

**Comments to the Author**

1. Is the manuscript technically sound, and do the data support the conclusions?

Reviewer #1: Yes

Reviewer #2: Partly

2. Has the statistical analysis been performed appropriately and rigorously? 

Reviewer #1: Yes

Reviewer #2: No

3. Have the authors made all data underlying the findings in their manuscript fully available?

Reviewer #1: Yes

Reviewer #2: No

4. Is the manuscript presented in an intelligible fashion and written in standard English?

Reviewer #1: Yes

Reviewer #2: Yes

5. Review Comments to the Author

Reviewer #1: Using eight plots and age-structure analysis, this study revealed that position of lower spruce treelines was retreating and tree recruitment rate was decreasing. The expansion of low-elevation forest has important implications for carbon sequestration and climate mitigation. Given that trees less than 1.3 m was not included in the plot survey, some uncertainties in tree recruitment data remained. Overall, results of this study is interesting to readers.

Specific comment:

Line 29: Some new references should be cited here. I recommend the treeline reference below.

Wang Y, Li D, Ren P, Ram Sigdel S, Camarero JJ. Heterogeneous responses of alpine treelines to climate warming across the Tibetan Plateau. Forests,2022,13:788.

Line 112-114: In Table 1, you only measured trees with height>1.3 m, so age of trees less than 1.3 m was not included in the data analysis.

Line 138: It is OK that you use the formula to calculate annual recruit rate. Since young trees less than 1.3 m was not included in the field plot, thus you should carefully consider the uncertainties in tree recruitment rate.

Line 178: Average value and SD (mean ±SD) should be shown in the fourth column.

Line 261-262: Position of lower treeline did not change between 2008-2018. Could you provide some explanation for this?

Line 317: delete

Reviewer #2: This study entitled ‘Fifty years of change in the lower tree line in an arid coniferous forest in the Qilian Mountains, northwestern China’ focused on the inverted treeline shifts in response to climate change. The inverted treeline is widely used to describe the grassland changes to forests. This manuscript is well written. However, there are some points need to be strictly phrased and seriously checked.

(1) During the period before 1998, forests in the lower elevational sites in Qilian Mountain are seriously affected by human activity such as clear cutting including the study sites. That is why the inverted treeline in plot 2 and 4 shift upward in 1968-1998. The select cutting may also result in the destruction of old trees thus causing a false appearance of inverted treeline decrease in their elevation.

(2) The authors only include the trees with height higher than 1.3 m. In fact, trees may take 15-20 years to reach 1.3 m in this area. The authors study the tree regeneration in response to climate change in the past fifty years. The exclusion of trees lower than 1.3 m may lead to serious bias to the results.

(3) There are numbers of studies focused on inverted treelines (e.g., Gilfedder et al. 1988). The formation of inverted nature treelines resulted from water logging, frost damage even in the arid region. It should be better to add more discussion. Also, comparing with the alpine treeline shifts are not reasonable to explain the treeline shifts.

6. PLOS authors have the option to publish the peer review history of their article (what does this mean?). If published, this will include your full peer review and any attached files.

Reviewer #1: No

Reviewer #2: No

---

## [Author Response · Author response to Decision Letter 0]

30 Aug 2023

Review 1:

Comment 1: Using eight plots and age-structure analysis, this study revealed that position of lower spruce treelines was retreating and tree recruitment rate was decreasing. The expansion of low-elevation forest has important implications for carbon sequestration and climate mitigation. Given that trees less than 1.3 m was not included in the plot survey, some uncertainties in tree recruitment data remained. Overall, results of this study is interesting to readers.

Response: Thank you for this comment. We have modified the language on the impact of ignoring the trees below 1.3m. For details, please refer to our reply to Comment 3 and Comment 4.

Comment 2: Line 29: Some new references should be cited here. I recommend the treeline reference below.

Wang Y, Li D, Ren P, Ram Sigdel S, Camarero JJ. Heterogeneous responses of alpine treelines to climate warming across the Tibetan Plateau. Forests,2022,13:788.

Response: Thank you for this suggestion. We have added some new literature on the tree line, including the one you mentioned.

Comment 3: Line 112-114: In Table 1, you only measured trees with height>1.3 m, so age of trees less than 1.3 m was not included in the data analysis. 

Comment 4: Line 138: It is OK that you use the formula to calculate annual recruit rate. Since young trees less than 1.3 m was not included in the field plot, thus you should carefully consider the uncertainties in tree recruitment rate.

Response: Thank you for this suggestion. Combining Comment 3 and Comment 4, we have the following answers and changes:

Initially, when designing the experiment, we considered that 1.3m meets the need of measuring breast diameter (height of breast diameter is 1.3m ), and also meets the requirement of tree height greater than 2m in the tree line area. After considering your comments, we consulted the relevant literature and found out that excluding trees below 1.3 m in height introduces great uncertainty about the results of tree recruitment data. Therefore, we abandoned the use of tree regeneration rates to explore the impact of climate change on the location of the lower tree line and modified the methods, results, and discussion, as applicable, of this aspect.

Changes in method：

To study the impacts of climate change on tree recruitment in the lower tree line, we used Pearson’s correlation coefficient between the average value of different meteorological factors in each decade and the mean distance of descent of the lower tree line tree between years 1968 and 2018 using SPSS 19. The meteorological factors we selected were annual average temperature (℃), annual precipitation (mm), spring average temperature (℃), summer average temperature (℃), autumn average temperature (℃), winter average temperature (℃), spring precipitation (mm), summer precipitation (mm), autumn precipitation (mm), and winter precipitation (mm).

Changes in results：

The correlation between the average value of different meteorological factors in each decade and mean distance of descent of the lower tree line between 1968 and 2018 is shown in Table 4. The results show that the change in the lower tree line was significantly correlated with the annual average temperature (℃) (r=0.907，P=0.033) and annual precipitation (mm) (r=0.911, P=0.031). The correlation between the degree of decrease in the tree line and the change in annual precipitation was the highest. In addition, the elevation of tree line decline was positively correlated with the change in summer temperature and the change in winter precipitation, with the correlation coefficient > 0.5. The elevation of decline in the tree line was negatively correlated with the changes in other meteorological factors, with the correlation coefficient of < 0.5.

Analysis of meteorological factors and changes in the location of the lower tree line showed that the trend in temperature warming in years 1968 to 2008 has been weakening, especially in 1998-2008 (the increase in temperature was < 1℃), and the lower tree line descent between 1968 and 2008 also slowed. There was a slight decrease in temperature after 2008, and the elevation of the lower tree line remained unchanged. Similarly, the trend of increasing precipitation in 2008-2018 has also weakened.

Table 4 The relationship between mean distance of descent elevation of the lower tree line (m) and different meteorological factors from 1968-2018. (M represents mean distance of descent elevation of the lower tree line (m), C1 to C10 represent annual average temperature (℃), annual precipitation (mm), spring average temperature (℃), summer average temperature (℃), autumn average temperature (℃), winter average temperature (℃), spring precipitation (mm), summer precipitation (mm), autumn precipitation (mm), winter precipitation (mm); * represents a significant correlation at the 0.05 level ( bilateral )) 

Correlation C1 C2 C3 C4 C5 C6 C7 C8 C9 C10

M R 0.907* 0.911* -0.352 0.703 -0.147 -0.243 -0.436 -0.288 -0.190 0.683 

 P 0.033 0.031 0.562 0.186 0.814 0.694 0.463 0.638 0.760 0.204 

Changes in discussion：

Recent studies have shown that precipitation has a comparable or stronger effect on the position of tree line than temperature, as enough precipitation and moisture may promote the growth rate of trees and the establishment of trees (58, 59).

Comment 5: Line 178: Average value and SD (mean ±SD) should be shown in the fourth column.

Response: Thank you for this suggestion. We added the Average value and SD (mean ±SD) at the bottom of the table. 

Table 2. Survey characteristics of the sites.

Plot Number Stand density（number/m2） Average DBH（cm） Standard deviation of DBH(cm) Average forest age（year） Standard deviation of forest age (year)

1 0.1 5.4 3.6 25 16 

2 0.26 6 6.8 27 29 

3 0.19 9.5 7.1 43 31 

4 0.12 8.5 6.6 38 29 

5 0.05 18 11.9 79 51 

6 0.07 13.2 12.1 58 52 

7 0.04 16.3 12.8 72 55 

8 0.11 12.3 7.2 55 31 

Comment 6: Line 261-262: Position of lower treeline did not change between 2008-2018. Could you provide some explanation for this?

Response: Thank you for this suggestion. We added this part in the discussion.

The correlation analysis between the change in the lower tree line position and meteorological factors showed that the change in the lower tree line was highly correlated with annual precipitation and annual average temperature; the rate of change in these two factors after 2008 has slowed down, so the elevation of decline in the lower tree line has also slowed down. At the same time, the change in trees in the tree line area may be from expansion of the forest to increase in tree density. During adaptation to climate change, forest ecosystems may fill in their interiors rather than shift the position of the tree line (52-54). In a study of the forest line in the Himalayan Mountains (including the Qilian Mountains), the changes in forest line exhibited high heterogeneity, with 67.7% of the tree lines shifting upward, 73% of the forest lines exhibiting an increase in tree recruitment, and 67.8% of the forest lines presenting acceleration of tree growth rates (4).

One survey showed that Picea crassifolia Kom. forest density increased 23-fold at the tree line, but the tree line position was not significantly altered during the past 100 years (55). Another survey indicated that the elevation of the upper tree line shifted upward by 6.1 to 10.4 m between 1957 and 1980, but showed no obvious changes in years 1980 to 2007 (33). Further, the number, area, and concentration of forest patches increased in years 1968 to 2017 in relatively flat and partly-sunny areas, but the rate of area increase and ascent of the tree line slowed after the year 2008 (56). 

Our results were similar to those of others from the Qilian Mountains; the lower tree line in the Qilian Mountains has shifted downslope in the past 50 years by an average of 9.82 m, and the expansion rate has begun to decline after 1998, with no change after 2008.

Comment 7: Line 317: delete

Response: Thank you for this suggestion. We deleted.

Review 2:

Comment 1: During the period before 1998, forests in the lower elevational sites in Qilian Mountain are seriously affected by human activity such as clear cutting including the study sites. That is why the inverted treeline in plot 2 and 4 shift upward in 1968-1998. The select cutting may also result in the destruction of old trees thus causing a false appearance of inverted treeline decrease in their elevation.

Response: Thank you for this suggestion. We have expanded the discussion of the causes of the change in the tree line.

Changes in the development of vegetation in the lower tree line are affected not only by periodic climate fluctuations and microenvironment caused by topography, but also by disturbances, especially the role of humans in fire frequency, forest coverage, or composition (66). In the Himalayas, human activity was one of the reason why the trees have not reached the height of the forest line height of the forest line that is limited by the average temperature (67). Before Qilian Mountain was identified as a national natural area in 1988, it was affected by human activities such as thinning (68). Therefore, the decline of tree line may be the illusion of the decline of tree line elevation caused by the destruction of old trees. 

Comment 2: The authors only include the trees with height higher than 1.3 m. In fact, trees may take 15-20 years to reach 1.3 m in this area. The authors study the tree regeneration in response to climate change in the past fifty years. The exclusion of trees lower than 1.3 m may lead to serious bias to the results.

Response: Thank you for pointing that out. we have the following answers and changes:

Initially, when designing the experiment, we considered that 1.3m meets the need of measuring breast diameter (height of breast diameter is 1.3m ), and also meets the requirement of tree height greater than 2m in the tree line area. After considering your comments, we consulted the relevant literature and found out that excluding trees below 1.3 m in height introduces great uncertainty about the results of tree recruitment data. Therefore, we abandoned the use of tree regeneration rates to explore the impact of climate change on the location of the lower tree line and modified the methods, results, and discussion, as applicable, of this aspect.

Changes in method：

To study the impacts of climate change on tree recruitment in the lower tree line, we used Pearson’s correlation coefficient between the average value of different meteorological factors in each decade and the mean distance of descent of the lower tree line tree between years 1968 and 2018 using SPSS 19. The meteorological factors we selected were annual average temperature (℃), annual precipitation (mm), spring average temperature (℃), summer average temperature (℃), autumn average temperature (℃), winter average temperature (℃), spring precipitation (mm), summer precipitation (mm), autumn precipitation (mm), and winter precipitation (mm).

Changes in results：

The correlation between the average value of different meteorological factors in each decade and mean distance of descent of the lower tree line between 1968 and 2018 is shown in Table 4. The results show that the change in the lower tree line was significantly correlated with the annual average temperature (℃) (r=0.907，P=0.033) and annual precipitation (mm) (r=0.911, P=0.031). The correlation between the degree of decrease in the tree line and the change in annual precipitation was the highest. In addition, the elevation of tree line decline was positively correlated with the change in summer temperature and the change in winter precipitation, with the correlation coefficient > 0.5. The elevation of decline in the tree line was negatively correlated with the changes in other meteorological factors, with the correlation coefficient of < 0.5.

Analysis of meteorological factors and changes in the location of the lower tree line showed that the trend in temperature warming in years 1968 to 2008 has been weakening, especially in 1998-2008 (the increase in temperature was < 1℃), and the lower tree line descent between 1968 and 2008 also slowed. There was a slight decrease in temperature after 2008, and the elevation of the lower tree line remained unchanged. Similarly, the trend of increasing precipitation in 2008-2018 has also weakened.

Table 4 The relationship between mean distance of descent elevation of the lower tree line (m) and different meteorological factors from 1968-2018. (M represents mean distance of descent elevation of the lower tree line (m), C1 to C10 represent annual average temperature (℃), annual precipitation (mm), spring average temperature (℃), summer average temperature (℃), autumn average temperature (℃), winter average temperature (℃), spring precipitation (mm), summer precipitation (mm), autumn precipitation (mm), winter precipitation (mm); * represents a significant correlation at the 0.05 level ( bilateral )) 

Correlation C1 C2 C3 C4 C5 C6 C7 C8 C9 C10

M R 0.907* 0.911* -0.352 0.703 -0.147 -0.243 -0.436 -0.288 -0.190 0.683 

 P 0.033 0.031 0.562 0.186 0.814 0.694 0.463 0.638 0.760 0.204 

Changes in discussion：

Recent studies have shown that precipitation has a comparable or stronger effect on the position of tree line than temperature, as enough precipitation and moisture may promote the growth rate of trees and the establishment of trees (58, 59).

Comment 3: There are numbers of studies focused on inverted treelines (e.g., Gilfedder et al. 1988). The formation of inverted nature treelines resulted from water logging, frost damage even in the arid region. It should be better to add more discussion. Also, comparing with the alpine treeline shifts are not reasonable to explain the treeline shifts.

Response: Thank you for pointing that out. We expanded our discussion to include new possibilities of the cause of the change in the lower tree line.

4.3 Uncertainty discussion

Tree line changes are complex and variable, with significant regional differences; therefore the reasons for such changes between the forest and the open ecosystem are multiple and still part of the debate. 

The annual mean temperature and annual precipitation are only two of several other meteorological factors that affect the change in the lower tree line. Our correlation analysis indicated that the summer temperature also had a positive correlation with the rate of descent of the forest line. The expansion of the lower tree line was related to the decrease in low temperatures or frosts damaging tree seedlings caused by the increase in summer minimum temperature (61). Lower nighttime minimum temperatures and short continuous frost-free periods may prevent the establishment of tree seedlings in the lower tree line through photoinhibition, tissue damage or frost heaving; fine soil may also hinder the establishment of tree seedlings in the lower tree line (62).

Topographic factors affecting local environmental conditions such as slope, can also affect the development of vegetation in the lower tree line (63). A study in a semi-arid area of Greece from 1945 to 2015 showed that changes in temperature and precipitation only have the theoretically achievable elevation of the tree line, the main factor determining whether the trees in the tree line area expanded was the difference in wind caused by topography (64). The main factors responsible for the distribution of Eucalyptus coccifera lower tree line on the Mt. Wellington Plateau, Tasmania, were soil drainage characteristics (65).

Changes in the development of vegetation in the lower tree line are affected not only by periodic climate fluctuations and microenvironment caused by topography, but also by disturbances, especially the role of humans in fire frequency, forest coverage, or composition (66). In the Himalayas, human activity was one of the reason why the trees have not reached the height of the forest line height of the forest line that is limited by the average temperature (67). Before Qilian Mountain was identified as a national natural area in 1988, it was affected by human activities such as thinning (68). Therefore, the decline of tree line may be the illusion of the decline of tree line elevation caused by the destruction of old trees. 

In addition, the position changes of the lower tree line in arid areas may also be due to the influence of tree regeneration, time lag of climate adaptation, and the disappearance of permafrost (69).

It is expected that the tree line in the eastern Himalayas will continue to expand in the end of the 21st century, resulting in the loss of at least 20 % of its existing habitat for endemic plants (70). Our study shows that, in addition to considering the increase of the elevation of the tree line, it is also necessary to consider the changes in the relationship between alpine plants brought about by the downward expansion of the tree line.

The growth of trees in the tree line area will gradually reduce the dependence on temperature increases, due in turn to local factors other than climate warming (71). Studies have shown using climate change was not a robust explanation of changes in the location of tree lines. More factors should be introduced to explore changes in forest lines so as to formulate more appropriate policies to cope with climate change.

---

## [Editor Report · Decision Letter 1]

7 Sep 2023

PONE-D-23-19267R1Fifty years of change in the lower tree line in an arid coniferous forest in the Qilian Mountains, northwestern ChinaPLOS ONE

Dear Dr. fang,

Thank you for submitting your manuscript to PLOS ONE. After careful consideration, we feel that it has merit but does not fully meet PLOS ONE’s publication criteria as it currently stands. Therefore, we invite you to submit a revised version of the manuscript that addresses the points raised during the review process.

We look forward to receiving your revised manuscript.

Kind regards,

Bao Yang, Ph.D, Prof.

Academic Editor

PLOS ONE

Journal Requirements:

Additional Editor Comments:

The authors dealt with the comments reasonably. However, some uncertainty remains and should be mentioned in the abstract. For example, human activity is an important factor influencing the treeline migration and should be indicated in the abstract. In addtion, since the authors use a short period 1968-2018 with a decadal resolution for analysis. I will be subjected to uncertainties due to a small number of degree of freedom.

---

## [Author Response · Author response to Decision Letter 1]

25 Sep 2023

Additional Editor Comments:

The authors dealt with the comments reasonably. However, some uncertainty remains and should be mentioned in the abstract. For example, human activity is an important factor influencing the treeline migration and should be indicated in the abstract. In addtion, since the authors use a short period 1968-2018 with a decadal resolution for analysis. I will be subjected to uncertainties due to a small number of degree of freedom.

Response: Thank you for this suggestion. We added the content of the effects of human activities on the tree line in the abstract. Additionally, we strengthened the discussion of the uncertainty caused by a relatively short temporal resolution and a small number of degree of freedom. 

Changes in abstract：

The change in the lower tree line was significantly correlated with the annual average temperature (℃) and annual precipitation (mm) and may be affected by human activities..

The part added in 4.3 

Picea crassifolia Kom. grows slowly, so we chose ten years as the time scale for studying tree line change. Then, to explore the relationship between tree line change and meteorological factors, the correlation analysis was done at that time scale, too. Therefore, the amount of data is small, and analysis of the relationship between tree line changes and meteorological factors supported the discussion on the possibility of tree line changes only. In follow-up work, tree rings will be considered to more accurately establish the relationship between tree growth and climate change.

---

## [Editor Report · Decision Letter 2]

27 Sep 2023

Fifty years of change in the lower tree line in an arid coniferous forest in the Qilian Mountains, northwestern China

PONE-D-23-19267R2

Dear Dr. fang,

We’re pleased to inform you that your manuscript has been judged scientifically suitable for publication and will be formally accepted for publication once it meets all outstanding technical requirements.

Kind regards,

Bao Yang, Ph.D, Prof.

Academic Editor

PLOS ONE

Additional Editor Comments (optional):

The authors have made the change. This manuscript is acceptable now.
---

## [Editor Report · Acceptance letter]

2 Oct 2023

PONE-D-23-19267R2 

Fifty years of change in the lower tree line in an arid coniferous forest in the Qilian Mountains, northwestern China 

Dear Dr. fang:

I'm pleased to inform you that your manuscript has been deemed suitable for publication in PLOS ONE. Congratulations! Your manuscript is now with our production department. 

Kind regards, 

on behalf of

Dr. Bao Yang 

Academic Editor

PLOS ONE